# NTKCPL: Active Learning on Top of Self-Supervised Model by Estimating True Coverage

## Abstract

High annotation cost has driven extensive research in active learning and self-supervised learning. Recent research has shown that in the context of supervised learning, when we have different numbers of labels, we need to apply different active learning strategies to ensure that it outperforms the random baseline. This number of annotations that change the suitable active learning strategy is called the phase transition point. We found, however, when combining active learning with self-supervised models to achieve improved performance, the phase transition point occurs earlier. It becomes challenging to determine which strategy should be used for previously unseen datasets. We argue that existing active learning algorithms are heavily influenced by the phase transition because the empirical risk over the entire active learning pool estimated by these algorithms is inaccurate and influenced by the number of labeled samples. To address this issue, we propose a novel active learning strategy, neural tangent kernel clustering-pseudo-labels (NTKCPL). It estimates empirical risk based on pseudo-labels and the model prediction with NTK approximation. We analyze the factors affecting this approximation error and design a pseudo-label clustering generation method to reduce the approximation error. Finally, our method was validated on five datasets, empirically demonstrating that it outperforms the baseline methods in most cases and is valid over a wider range of training budgets.

## 1 Introduction

The boom in deep learning models in recent years stems in part from the massive amounts of data [11, 17, 23]. However, the demand for large amounts of data, especially labeled data, in turn, constrains the application of deep learning models, since large amounts of labels imply high annotation costs [41, 1, 45]. Active learning is a path to alleviate the cost of labeling by selecting informative subsets of samples to annotate.

However, the benefits of active learning have been increasingly questioned in recent years [25, 28]. One of the main concerns is that training a model initialized by self-supervised learning with randomly selected labeled samples often yields results far beyond those obtained by existing active learning with supervised training (randomly initialized or initialized by the last round of the active learning model) [6, 8, 7, 14, 9]. Because the latter only uses labeled data to train the network, while the former uses a large amount of unlabeled data to train the backbone of the network. Since most existing active learning algorithms are designed in the context of supervised training, they must be validated with a large number of labels compared to the number of labels required in training from a self-supervised model. This means that the effectiveness of these active learning algorithms is not guaranteed in the case of having access to relatively few annotations, as is the case when combining with a self-supervised model. Several studies [15, 42, 4] have shown that many existing active

learning strategies fail to outperform the random baseline when combining them with self-supervised learning. In this paper, we focus on designing an active learning strategy that works well in the training method with a self-supervised model.

The "phase transition" phenomenon [15] is known to occur in active learning with supervised training. It refers to the fact that an active learning strategy that outperforms a random baseline when the total number of labels is small will be inferior to a random baseline when the total number of labels is large (called the **low-budget** strategy) and vice versa (called the **high-budget** strategy). We note that when combining active learning with the self-supervised model, the cut-off point between low-budget and high-budget strategy occurs much earlier. For example, in the CIFAR-100 [21], the cut-off point is about 10,000 labeled samples when training in the supervised learning way [16]. But, the cut-off point shifts forward to about 1,500 labeled samples when training from a self-supervised model. The forward-moving cut-off point means that even if the annotation budget is low (only one order of magnitude above the number of classes in the dataset), it is likely to hit that cut-off point. Thus, for a previously unseen dataset, it is difficult to simply determine whether a low-budget or high-budget strategy should be chosen since the difficulty varies from dataset to dataset. In this paper we use this problem to motivate the design of an active learning strategy with a wider effective budget range.

Since existing low-budget strategies are designed based on the idea of feature space coverage [24, 15, 42], we first analyze the problems of determining coverage based on sample feature distances in sec. 2. After that, we propose that the true coverage where the empirical risk is zero, can be estimated based on pseudo-labels and predictions of the model trained on the candidate set. Based on this, we propose our active learning strategy, Neural Tangent Kernel Clustering-Pseudo-Labels (NTKCPL), which uses the NTK [18, 27] and CPL to approximate empirical risk on active learning pool in sec. 3.2. And we analyze which factor affects approximation error in sec. 3.3. Based on this analysis, we design a CPL generation method in sec. 3.4. Extensive experimental results demonstrate that our method outperforms state-of-the-art approaches in most cases and has a wider effective budget range. As part of the results (sec. 4) we also show our method is effective for self-supervised features of different quality.

Our contribution is summarized as follows: (1) We propose a novel active learning strategy, NTKCPL, by estimating empirical risk on the whole active learning pool based on pseudo-labels. (2) We analyze the approximation error of the empirical risk in the active learning pool when NTK and CPL are used to approximate networks and true labels. (3) Our method outperforms both low- and high-budget active learning strategies within a range of annotation quantities one order of magnitude larger than traditional low-budget active learning experiments. This means that our approach can be used more confidently for active learning on top of self-supervised models than existing low-budget strategies.

## 1.1 Related Work

Most active learning strategies are designed and validated in the high-budget scenario where network weights are randomly initialized or initialized from the weights of the previous active learning round. Active learning methods mainly include uncertainty-based sampling [22, 13, 19], feature space coverage [32, 24, 42, 33, 5, 40], the combination of uncertainty and diversity [41, 3], learning-based methods [43], and so on [34, 35]. Moreover, some recent studies explore **"look ahead"** strategies [26, 38], where samples are selected based on the model trained on candidate training sets. However, with the development of self-supervised training, the training approach for low-budget scenarios has shifted to training based on a self-supervised pre-trained model [24]. This change in the training method implies a shift in the total number of samples that need to be selected by active learning. When training based on a self-supervised model, often only 0.4-6% of the total data needs to be labeled to achieve similar results to training with 20-40% labeled data on a randomly initialized network [4]. Recent studies have shown that there exists a phase transition phenomenon in active learning strategies, whereby opposite strategies should be adopted in high-budget and low-budget scenarios [15], causing many active learning strategies designed for high-budget scenarios unsuitable for training based on a self-supervised model. As a result, recent studies have explored active learning strategies specifically designed for low-budget scenarios [15, 42, 31, 20]. However, we find that these strategies are effective only when the number of labeled data samples is extremely small, and as we increase the labeled data to one order of magnitude above the number of classes of the dataset, their performance falls below that of the random baseline.

## 2 Insight: Distance is not an accurate indicator of empirical risk

The goal of the active learning is to find a labeled subset, $D_C = (x_i, y_i)_{i=1}^{N_C}$, such that the model trained on that subset, $f_{D_C}$, has the minimized empirical risk in the entire active learning pool, $D = (x_i, y_i)_{i=1}^{N}$ as shown in eq. 1.

$$argmin_{D_C} \frac{1}{N} \sum_{i \in D} Loss(f_{D_C}(x_i), y_i) \tag{1}$$

Unfortunately, during active learning, we do not have the labels of the entire active learning pool, so we cannot compute this loss directly. To address this problem, current methods [32, 24, 33] covert empirical risk minimization into feature space coverage based on Lipschitz continuity. Although Lipschitz continuity guarantees that the difference between the model's predictions is less than the product of the Lipschitz constant and the difference between inputs, it does not guarantee that their predictions fall into the same class. In practice, we cannot determine the true coverage because we do not know the distance threshold beyond which the model would change its predicted class for unlabeled samples.

Therefore, the current solution is to minimize the coverage radius assuming full coverage [32] or to maximize coverage based on high purity coverage [42], where purity refers to the probability that the sample has the same label within a given distance. Assuming full coverage leads to an overestimated coverage as shown in fig. 1a, i.e., some covered samples still have a large empirical risk, while high-purity coverage causes underestimated coverage as shown in fig. 1b. The overestimated coverage may cause the active learning algorithm to miss samples in areas that are not truly covered, while underestimated coverage makes active learning algorithms likely to select redundant samples. These affect the performance of active learning.

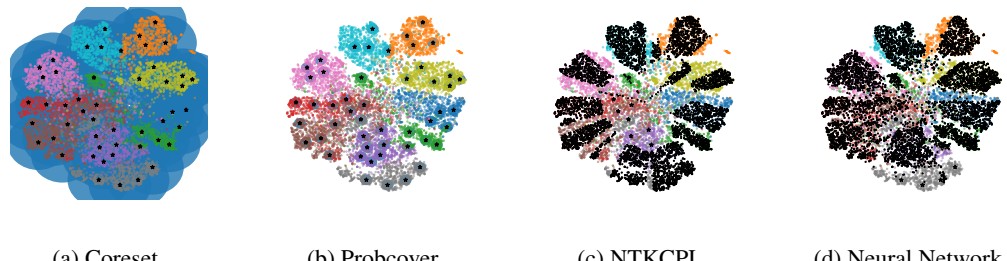

| (a) Coreset | (b) Probcover | (c) NTKCPL | (d) Neural Network |

Figure 1: Coverage estimation based on sample feature distance vs. NTKCPL. Here different colors represent different categories, the black star denotes labeled samples and the blue circle represents the samples considered covered based on the feature distance approach. Coreset assumes full coverage and Probcover assumes high purity coverage. The coverage estimated by our method, NTKCPL, and true coverage based on predictions of the neural network is represented by black dots. The coverage estimated by NTKCPL is more consistent with the true coverage of the neural network than those estimated based on feature distances.

Additionally, estimating the empirical risk based on distance implies the assumption that model predictions are only relevant to the nearest labeled sample, which is often not the case in reality. To estimate the true coverage, we propose a new strategy, NTKCPL. It estimates the empirical risk based on the predictions of the model trained on the candidate set and pseudo-labels.

## 3 Method: NTKCPL

In sec. 3.1, we briefly review the Neural Tangent Kernel (NTK) [18] that enables active learning strategies based on the outputs of a model trained on a candidate set feasible. Then, we propose our active learning strategy, NTKCPL, in sec. 3.2 and analyze the approximation error of NTKCPL in sec. 3.3. Finally, based on the analysis, we introduce the method of generating cluster pseudo-label in sec. 3.4.

## 3.1 Preliminaries

Neural Tangent Kernel (NTK) is a powerful tool to analyze the training dynamics of neural network. Jacot et al. [18] show that the neural network is equivalent to the kernel regression with Neural Tangent Kernel when network is sufficiently wide and its weights are initialized properly [2]. The NTK, $\mathcal{K}$, is shown in eq. 2, where the $f$ denotes a neural network with parameters $\theta$ and $\mathcal{X}$ denotes train samples. When training with MSE loss, the neural network has a closed-form solution for the prediction of test sample $x$ at iteration $t$ as eq. 3, where $\mathcal{Y}$ denotes labels of trainset and $f_0$ denotes the output of network with initialized weights.

$$\mathcal{K}(\mathcal{X}, \mathcal{X}) = \nabla_\theta f(\mathcal{X}) \nabla_\theta f(\mathcal{X})^T \tag{2}$$

$$f_t(x) = f_0(x) + \mathcal{K}(x, \mathcal{X})\mathcal{K}(\mathcal{X}, \mathcal{X})^{-1}(\mathcal{I} - e^{-t\mathcal{K}(\mathcal{X}, \mathcal{X})})(\mathcal{Y} - f_0(\mathcal{X})), \tag{3}$$

Additionally, for active learning scenarios, Mohamad [26, 27] proposes the computation time of using NTK can be further reduced by considering the block structure of the matrix, which means that look ahead type active learning strategies can be implemented in a reasonable amount of time. For example, as shown in [26], if we want to use the look ahead active learning strategy, each active learning cycle takes 3 hours to train the entire network of 15 epochs on the MNIST dataset, while it takes only 3 minutes to use NTK with a block structure.

## 3.2 Framework

We propose a look ahead strategy, NTKCPL, to approximate the empirical risk on the whole active learning pool directly. There are two challenges: (1) estimate empirical risk without labels and (2) estimate predictions of models trained with candidate sets efficiently and accurately.

For the first challenge, clusters on self-supervised features provide good pseudo-labels. Because most samples in the same cluster have the same label [39]. And when the number of clusters is increased, it can improve the purity of clusters, where purity refers to the probability that the sample has the same label within the same cluster. We call these clusters clustering-pseudo-labels (CPL), $y_{cpl}$.

For the second challenge, as introduced in sec. 3.1, NTK approximates the network well for random initialization and the computation time is acceptable. However, in our scenario, training on top of the self-supervised model, NTK does not approximate predictions of the whole network well. The main reason is that weights of the neural network are initialized by self-supervised learning rather than NTK initialization, i.e., drawn i.i.d. from a standard Gaussian [18]. In addition, the self-supervised initialization provides the neural network with a powerful feature representation capability that is not available in NTK. This leads to inconsistency between NTK predictions and network outputs. So, in our method, the NTK is used to approximate the classifier instead of the whole network. And the inputs of NTK are self-supervised features. Accordingly, we choose a training method following [24] that freezes the encoder initialized by self-supervised learning and trains only the MLP as a classifier. That training method achieves better or equal performance than fine-tuning the whole network in the low-budget case while its prediction is more consistent with the results of NTK. We denotes the predictions of NTK with trainset $D_C$ as $\hat{f}_{D_C}$. Now, the active learning goal in eq. 1 is approximated as eq. 4.

$$argmin_{D_C} \frac{1}{N} \sum_{i \in D} Loss(\hat{f}_{D_C}(x_i), y_{cpl,i}) \tag{4}$$

The algorithm is shown in Alg. 1. For computational simplicity and without loss of generality, we use 0-1 loss to calculate empirical risk in eq. 4. In each round of active learning, after computation of NTK based on eq. 2 and generation of CPL based on the method introduced in sec. 3.4, the sample that minimizes the empirical risk on the whole active learning pool after adding labeled set is selected.

## 3.3 NTKCPL Approximate Error

In this section, we analyze what affects the accuracy of NTKCPL estimates of empirical risk on the whole active learning pool. The difference between the true empirical risk and the estimated

**Algorithm 1** NTKCPL

---

1: **Input:** self-supervised feature $f_{self}$, active learning feature $f_{al}$ labeled set $L$, unlabeled set $U$, budget $b$, initial budget $b_0$, maximum cluster number $C_{max}$, model prediction $Y_{pre,t-1}$ at the last active learning round
2: **Output:** labeled set $L$, model prediction $Y_{pre,t}$ at this round
3: **if** $L$ is $\emptyset$ **then**
4:      $Y_{cpl} \leftarrow$ K-means$(f_{self}, b_0)$
5: **else**
6:      $N_{clu} = min\{b_i/2, C_{max}\}$
7:      $Y_{cpl} \leftarrow$ CPL generation$(f_{al}, Y_{pre,t-1}, b_0, N_{clu}, L)$ based on Alg. 2
8: **end if**
9: Initialize classifier, MLP, compute $f_0$ and $ker$ based on eq. 2
10: **for** $itr = 1$ **to** $b$ **do**
11:      $Emp\_risk = []$
12:      **for** $(x_i, y_{cpl,i})$ **in** $U$ **do**
13:          Compute $Y_{NTK} = \hat{f}(ker, f_0, L \cup (x_i, y_{cpl,i}), U)$ based on eq. 3
14:          $Emp\_risk += [\text{0-1}Loss(Y_{NTK}, Y_{cpl})]$
15:      **end for**
16:      $i' = argmin Emp\_risk$
17:      $L = L \cup (x_{i'}, y_{cpl,i'}), U = U \backslash x_{i'}$
18: **end for**
19: Query label $y_{i'_{1,\dots,b}}$ of $x_{i'_{1,\dots,b}}$
20: $L = L \cup (x_{i'_{1,\dots,b}}, y_{i'_{1,\dots,b}}), U = U \backslash x_{i'_{1,\dots,b}}$
21: Train classifier $f_t$ on $L$
22: model prediction $Y_{pre,t} = f_t(U)$

---

empirical risk using NTK and CPL is shown in eq. 5. The approximation error can be divided into two terms, the first one is the difference between NTK and neural network prediction, $error_{NTK}$, and the second one is the difference caused by CPL during NTK estimation, $error_{CPL}$. For the $error_{NTK}$, as we mentioned in sec. 3.2, NTK is used to approximate the classifier only to obtain better consistency. To analyze $error_{CPL}$, we start with the relationship between the predictions of NTK trained with the ground truth, $\hat{f}_y(x_i)$, and CPL, $\hat{f}_{cpl}(x_i)$.

$$\frac{1}{N}\sum_{i \in D}\left|Loss(f(x_i), y_i) - Loss(\hat{f}(x_i), y_{cpl,i})\right|$$
$$\leq \quad \frac{1}{N}\sum_{i \in D}\left(\left|Loss(f(x_i), y_i) - Loss(\hat{f}(x_i), y_i)\right| + \left|Loss(\hat{f}(x_i), y_i) - Loss(\hat{f}(x_i), y_{cpl,i})\right|\right) \quad (5)$$

**Definition** Denotes the $j^{th}$ output of $\hat{f}_{cpl}$ as $\hat{f}_{cpl}^j$. Label mapping function $g$ converts NTK's predictions about CPL classes, $\hat{f}_{cpl}(x_i)$, into predictions about true classes, $\hat{f}_{ymap}(x_i)$, based on dominant labels within corresponding CPL classes as shown in eq. 6, where $D_{dom}$ is a set of index $k$, where $j$ is the dominant true label classes within CPL class, $y_{cpl,k}$.

$$\hat{f}_{ymap}^j(x_i) = \sum_{k \in D_{dom}} \hat{f}_{cpl}^k(x_i) \quad (6)$$

**Proposition** If the true labels of labeled samples are the dominant labels in their corresponding CPL clusters, $\hat{f}_y(x_i) = g(\hat{f}_{cpl}(x_i))$. We defer the proof to appendix 1.

$$error_{CPL} = P_{nff} + P_{fnf} \quad (7)$$

As mentioned in sec. 3.2, we use 0-1 loss to calculate empirical risk. We can expand $error_{CPL}$ as eq. 7, where we denote the probability that the NTK prediction agrees with the $y$ but not with $y_{cpl}$ as $P_{fnf}$, and the probability that the NTK prediction does not agree with $y$ but agrees with $y_{cpl}$ as $P_{nff}$. According to the proposition, we argue $argmax \hat{f}_y(x_i)$ is most likely equal to $g(argmax \hat{f}_{cpl}(x_i))$.

$P_{fnf}$ refers to the case where different CPL classes correspond to the same true label class, i.e., over-clustering. $P_{nff}$ means that the true label of a sample is different from the dominant true label within its CPL class, i.e., the CPL class includes samples from different true label classes, which is called impurity. Detailed explanations and empirical evidence can be found in appendix 1.

### 3.4 Cluster Pseudo-Labels

As shown by eq. 7, the effect of CPL on the approximation error comes from the purity of the clusters and over-clustering. To improve clustering purity, we take two approaches: (1) clustering on the active learning feature, i.e., the output of the penultimate layer of the classifier, and (2) increasing the number of clusters. However, increasing the number of clusters may cause the labeled samples not to cover all classes of the CPL (under-coverage) and also increase the over-clustering error. For example, a group of samples with the same true label is clustered into $K$ different classes. Even though NTK incorrectly predicts some samples as other CPL classes, their true empirical risk is zero.

To improve the under-coverage, we set the number of clusters to half of the total number of labels, i.e., each cluster includes two labeled samples on average. To improve the over-clustering, we manually set the maximum number of clusters and design a clustering-splitting approach instead of directly increasing the number of clusters. It splits the low-purity clusters and keeps the high-purity ones to reduce the extra over-clustering errors within samples located in the high-purity clusters. Specifically, we use the prediction of the neural network in each round of active learning to estimate the number of confusing samples within each cluster, i.e., the number of samples from classes that are different from the dominant class. The clusters that contain the

---

**Algorithm 2** CPL generation

---

**Input:** active learning feature $f_{al}$, model predictions $Y_{pre}$, initial cluster number $C_0$, cluster number $C_{max}$, labeled set $L$
**Output:** CPL $Y_{cpl}$
$Clu_{1,...,C_0} \leftarrow$ Constrained K-means($f_{al}, L, C_0$)
**for** $itr = 1$ **to** $(C_{max} - C_0)$ **do**
    $i' = argmax_i$ number of Confusing samples($Clu_i$, $Y_{pre}$)
    $f_{al,i'} \leftarrow f_{al}$ of samples within $Clu_{i'}$
    $Clu_{i',C_0+1} \leftarrow$ K-means($f_{al,i'}, 2$ )
    $C_0 \leftarrow C_0 + 1$
**end for**
$Y_{cpl} \leftarrow Clu_{1,...,C_{max}}$

---

largest number of confusing samples are split sequentially until a predefined number of clusters is reached. The cluster splitting algorithm is shown in Alg. 2, where we adopt the constrained K-Means [37] to improve the clusters from labeled sample constraints.

## 4 Experiment Results

Our approach is validated on five datasets with various qualities of self-supervised features. Datasets with good self-supervised features, such as CIFAR-10 [21], CIFAR-100 [21], and ImageNet-100 (a subset of ImageNet [11], following splitting in [36]), are included. SVHN [29] with poor self-supervised features is also included. Additionally, we consider practical scenarios where the total number of samples in the trainset is insufficient to support effective self-supervised training, such as Oxford-IIIT Pet dataset [30]. In this case, we evaluated the effectiveness of our method based on the model pre-trained on ImageNet [11].

**Baseline** We compare our proposed method with representative active learning strategies: (1) Random, (2) Entropy (uncertainty sampling, maximum entropy of output) [22], (3) Coreset (diversity active learning strategy, greedy solution of minimum coverage radius) [32], (4) BADGE (combination of uncertainty and diversity, kmeans++ sampling on grad embedding) [3], where the scalable version [10, 12], badge partition, is used in ImageNet-100, CIFAR-100 and Oxford-IIIT Pet because the huge dimension of grad embedding (5) Typiclust (designed for low-budget case) [15], (6) Lookahead (maximum output change based on NTK) [26].

**Implementation** Our method focuses on the low-budget regime, we followed the training method in [24], freezing weights of backbone initialized with self-supervised learning and then training a MLP as the classifier. The hyperparameters for training are set following [15] and can be found in appendix 3. For the self-supervised model, we adopt simsiam [9] for CIFAR-10, CIFAR-100 and SVHN and BYOL [14] for ImageNet-100 and Oxford-IIIT Pet. Resnet-18 [17] is used in CIFAR-10

Table 1: Comparison of accuracy of different active learning strategies on CIFAR-10. All results are averages over 5 runs. The best results are shown in red and the second-best results are shown in blue.

| #Labels | Random | Entropy | Coreset(self) | BADGE | TypiClust | LookAhead | NTKCPL(self) | NTKCPL(al) |
|---|---|---|---|---|---|---|---|---|
| 20 | 41.80±3.82 | 38.58±2.86 | 20.08±2.75 | 39.85±3.91 | 46.38±1.61 | 40.93±4.04 | 54.31±3.74 | 52.67±3.70 |
| 40 | 57.52±3.34 | 51.10±4.21 | 36.67±6.29 | 54.99±3.43 | 66.18±2.45 | 58.55±2.71 | 68.60±2.50 | 63.55±2.89 |
| 60 | 65.88±3.07 | 64.46±3.42 | 46.39±7.41 | 65.23±1.40 | 72.93±1.77 | 66.96±2.90 | 75.09±1.69 | 72.22±2.11 |
| 80 | 69.35±3.31 | 70.49±3.05 | 58.96±6.15 | 70.76±1.86 | 76.98±1.04 | 72.71±1.94 | 78.51±1.61 | 75.32±0.92 |
| 100 | 74.11±1.16 | 74.34±1.92 | 62.64±5.07 | 75.40±0.99 | 78.24±1.28 | 75.97±2.04 | 80.30±1.17 | 78.45±1.19 |
| 200 | 80.90±0.90 | 79.86±1.77 | 76.93±3.56 | 82.20±1.14 | 83.16±0.61 | 81.89±1.31 | 83.77±1.04 | 81.87±1.02 |
| 300 | 82.80±0.93 | 81.43±2.23 | 82.64±1.42 | 84.53±0.46 | 84.16±0.25 | 83.29±0.89 | 85.00±0.54 | 83.78±1.05 |
| 400 | 84.04±0.49 | 83.37±1.31 | 84.56±1.15 | 84.75±0.40 | 85.13±0.27 | 84.59±0.59 | 85.64±0.38 | 84.73±0.85 |
| 500 | 84.97±0.78 | 84.24±0.89 | 85.23±0.59 | 85.57±0.51 | 85.37±0.15 | 85.31±0.12 | 85.72±0.22 | 85.48±0.65 |
| 1000 | 86.26±0.38 | 84.94±0.48 | 86.75±0.36 | 86.06±0.31 | 86.07±0.14 | 85.69±0.47 | 86.83±0.33 | 87.15±0.57 |
| 1500 | 86.95±0.27 | 85.85±0.39 | 87.03±0.13 | 87.05±0.36 | 86.37±0.11 | 86.82±0.23 | 87.18±0.41 | 87.58±0.29 |
| 2000 | 87.30±0.37 | 86.92±0.15 | 87.34±0.27 | 87.31±0.47 | 86.55±0.21 | 87.16±0.19 | 87.34±0.41 | 87.87±0.39 |

and SVHN, WRN28-8 [44] is used in CIFAR-100 and Resnet-50 [17] is used in ImageNet-100 and Oxford-IIIT Pet.

The number of clusters in our method is set according to three rules, in the initial selection, it is set to the number of query samples, after that it is set to half of the query samples until the number of clusters reaches the maximum number of clusters. For CIFAR-10, CIFAR-100, ImageNet-100, SVHN, and Oxford-IIIT Pet, the maximum number of clusters is 100, 500, 300, 100, and 150, respectively. We followed [26] to sample a subset of the unlabeled set as the candidate set to select samples and estimate coverage. The candidate set includes 10,000 samples.

For the query step, most of the experiments (those on CIFAR-100, SVHN and Oxford-IIIT Pet) following the active learning literature by drawing a fixed number of samples from the unlabeled dataset to the oracle. Specifically, 500 for CIFAR-100, 20 for SVHN, and 40 for Oxford-IIIT Pet. We empirically found that fixed active learning query steps lead to much faster growth of classifier accuracy in the early stages of active learning (the amount of labels is about 10 times than the number of class) than in the later stages, so it is difficult to clearly observe the differences between different active learning strategies. For this reason, we empirically set varying query steps in our experiments with CIFAR-10 and ImageNet-100. Smaller query steps were used in the early stage of active learning and switched to larger query steps in the later stage. Specifically, for CIFAR-10, 20 samples are queried before 100 labels are available, 100 samples are queried before 500 labels and 500 labels are queried before 2000 labels. For ImageNet-100, 200 samples are selected before 1000 labels are available and 500 samples are queried before 2000 labels.

## 4.1 Main Results

All experiments were run 5 times and the avg. and std. are reported. Considering that the experiments are conducted for the scenario with a low annotation budget, it is not practical to construct a validation set to select the best checkpoints (the benefits of constructing a validation set are much less than using these labeled samples as training samples). Therefore, we report the final checkpoint accuracy, not the accuracy of the checkpoints determined by the validation set. The results are shown in fig. 2 and table 1. The detailed results are in appendix 5.

**NTKCPL outperforms SOTA.** As shown in table 1, fig. 2. In most cases, our proposed method outperforms the baseline methods. For the few cases with only a small number of labels, our method shows comparable performance with the low-budget strategy, TypiClust, such as in CIFAR-100 with 500 and 1000 labeled samples, and Oxford-IIIT Pet with 80 and 100 labeled samples.

**NTKCPL still shows good performance when the self-supervised features do not correspond well to the label classes.** Since the loss of self-supervised training is different from that of image classification, self-supervised features do not always correspond well with label classes. In SVHN dataset, self-supervised features of different classes are mixed together because the images include some irrelevant digits on both sides of the digit of interest [29]. Our method is similar to other

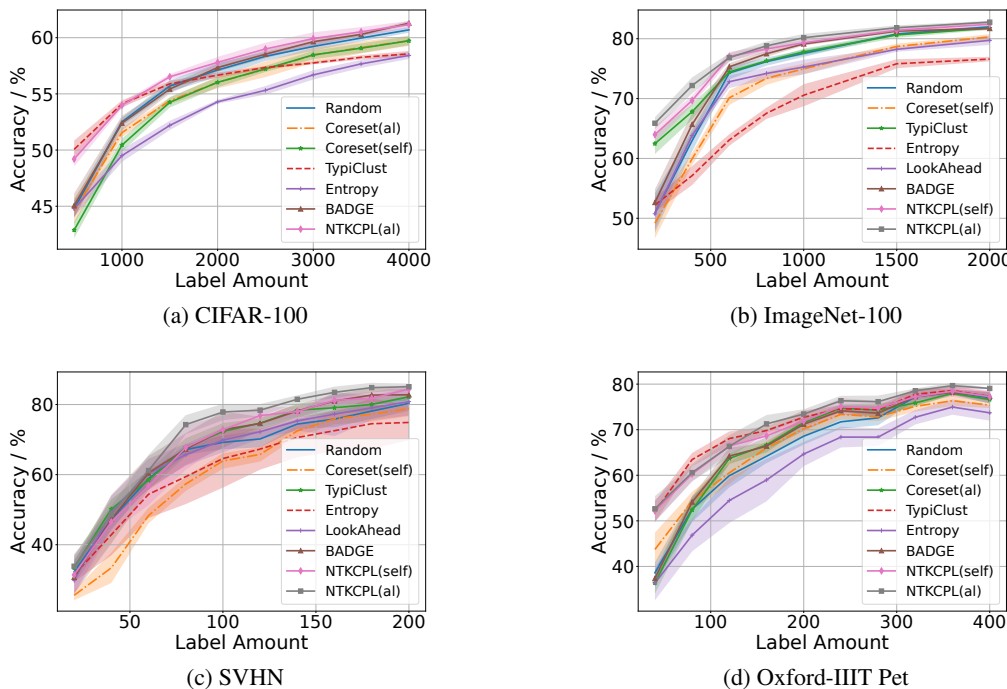

Figure 2: Performance of different active learning strategies. The shaded area represents std.

baseline strategies at the beginning of active learning, but it shows better results than baselines after several active learning rounds as shown in fig. 2c.

Another common scenario is the lack of sufficient samples to support effective self-supervised training. To evaluate in this context, we choose the Oxford-IIIT Pet dataset with the self-supervised model trained on ImageNet. The result is shown in fig. 2d. Our method has similar accuracy in the first three rounds as the TypiClust and outperforms all baseline methods afterward.

**NTKCPL has a wider effective budget range than SOTA.** Active learning based on self-supervised models exhibits an intensified phase transition phenomenon. We plot the active learning gain of our method and baselines on different datasets in fig. 3. The average accuracy of our method, NTKCPL(al), outperforms the random baseline at all quantities of labels. In contrast, both the typical high-budget strategy, BADGE, and low-budget strategy, TypiClust, appear to be worse than the random baseline over a range of annotation quantities. We show

Table 2: Comparison of the effective budget ratio of different active learning strategies.

|  | Effective Budget Ratio |
| --- | --- |
| TypiClust | 40.8% |
| BADGE | 42.0% |
| NTKCPL(al) | 92.7% |

the effective budget range of our method, NTKCPL, as well as the typical high-budget strategy, BADGE, and the typical low-budget strategy, TypiClust, across all experiments in table 2. The effective budget ratio refers to the proportion of the effective annotation quantity to the total annotation quantity, where the effective annotation quantity refers to the number of annotations at which active learning accuracy exceeds the random baseline (avg. + std.).

## 4.2 Ablation Study

In this section, we evaluate the coverage estimation of our method and the effect of the maximum cluster number on NTKCPL. Also, we compare the effect of generating CPL on self-supervised features as well as on the active learning feature on the performance of NTKCPL.

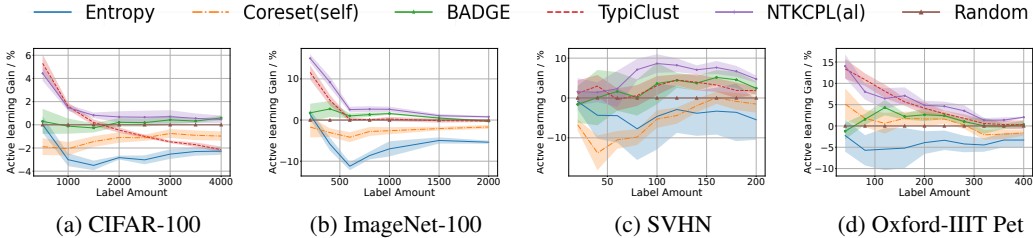

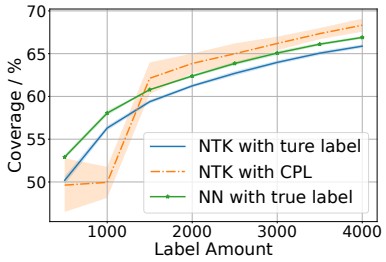
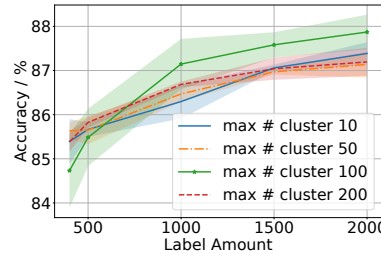

Figure 3: Active learning gain of different active learning strategies.

Figure 4: Coverage estimation on CIFAR-100.

Figure 5: Effect of the maximum number of clusters on active learning performance on CIFAR-10.

**Coverage Estimation** We conducted experiments on CIFAR-100, where the coverage indicates the proportion of samples that are correctly predicted. The estimated coverage of NTK with true label and with CPL is shown in fig. 4. Our method approximates the true coverage well for most cases.

**Effect of the Maximum Number of CPL** The ablation experiments are conducted on CIFAR-10. We plot the accuracy when the number of annotations selected by active learning is greater than 400 as shown in fig. 5. In this range, the number of classes of CPL is fixed at 10, 50, 100, and 200, respectively. The experimental results support our analysis in sec. 3.4 that too many or too few clusters will increase the approximation error, which affects the performance of active learning.

**Effect of self-supervised feature-based and active learning feature-based clustering-pseudo-labels on NTKCPL.** We denote NTKCPL based on active learning features as NTKCPL(al) and NTKCPL based on self-supervised learning feature as NTKCPL(self). The results are shown in table 1 and fig. 2. From these experiments, we found that clustering on active learning features yields better results except for the case where the number of annotations is very small. Also, NTKCPL(self) is better than NTKCPL(al) in a wide range of annotation quantities (no more than 500), when self-supervised features are good such as experiment in the CIFAR-10.

## 5 Conclusion

We study the active learning problem when training on top of a self-supervised model. In this case, an intensified phase transition is observed and it influences the application of active learning. We propose NTKCPL that approximates empirical risk on the whole pool more directly. We also analyze the approximation error and design a CPL generation method based on the analysis to reduce the approximation error. Our method outperforms SOTA in most cases and has a wider effective budget range. The comprehensive experiments show that our method can work well on self-supervised features with different qualities.

Our approach is limited to the fixed training approach, i.e., training the classifier on top of a frozen self-supervised training encoder, which is restricted to the low-budget scenario because the fine-tuning training approach provides higher accuracy in the high-budget case. Therefore, (1) how to accurately approximate the fine-tuning model initialized with self-supervised weights using NTK and (2) whether the samples selected by our current method have good transferability for the fine-tuning would be interesting future directions.

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
