# OpenReview forum: "NTKCPL: Active Learning on Top of Self-Supervised Model by Estimating True Coverage"
_NeurIPS.cc/2023/Conference — Submitted to NeurIPS 2023_

### Official Review · Reviewer_VAm9 · 2023-07-02

**Soundness:** 3 good
**Presentation:** 3 good
**Contribution:** 3 good
**Rating:** 7
**Confidence:** 2

**Summary:**

This paper proposed a new NTKCPL method to reduce the approximation error and it has a wider effective budget range in the setting of active learning on top of self-supervised model. The experimental results on several Computer Vision datasets (e.g., CIFAR-10, CIFAR-10, SVHN) validate the effectiveness of the proposed methods.

**Strengths:**

1. This paper is well-written. The problem this paper focuses on is important, and the proposed method is interesting.
2. This paper provides both theoretical and empirical results, which is great for a top machine learning conference like NeurIPS. The experiments are sufficient, and the conclusion is convincing.


**Weaknesses:**

1. A case study is suggested. For example, in CIFAR-10 dataset, which class or classes are much better than others or all the classes become better with the same scale? Providing detailed case study about the datasets together with improved metrics will make your conclusion further convincing.

**Questions:**

1. Did authors try this method in NLP datasets? It will be very great if the proposed method can be evaluated in different fields. In NLP, pre-trained model seems to play a more important role than the one in Computer Vision tasks.

**Limitations:**

Some potential limitations are suggested to add. For example, a better approximation method may need more computational resources.

---

> ### Author Rebuttal · Authors · 2023-08-10
>
> Thanks so much for your constructive reviews.
>
> **For weakness 1:**
>
> Thank you for your suggestion. We incorporated your feedback and added a new table to showcase the class-wise accuracy of our active learning strategy on the CIFAR-10 dataset. With the exception of classes 3 and 5 (true labels: cat and dog), which are often confused in the self-supervised feature space, our method generally demonstrates improved accuracy across the other classes. In addition, we will include a case study in the revised version of the paper.
>
> | Sample selection method | # labels | Class0 | Class1 | Class2 | Class3 | Class4 | Class5 | Class6 | Class7 | Class8 | Class9 | Avg. |
> | ----------- | ----------- | ----------- | ----------- | ----------- | ----------- | ----------- | ----------- | ----------- | ----------- | ----------- | ----------- | ----------- |
> | Random | 20 | 51.03 | 60.10 | 56.53 | 28.90 | 33.73 | 31.03 | 42.70 | 23.27 | 33.73 | 50.63 | 41.17 |
> | TypiClust | 20 | 40.63 | 55.03 | 63.67 | 43.83 | 53.30 | 30.70 | 57.83 | 50.57 | 37.47 | 34.80 | 46.78 |
> | NTKCPL(self) | 20 | 44.57 | 59.77 | 47.40 | 27.77 | 63.07 | 51.57 | 57.87 | 55.03 | 60.67 | 73.27 | 54.09 |
> | Random  |  100  |  65.70  |  89.37 |  59.67  |  56.87  |  65.83  |  66.07  |  85.33  |  73.00  |  91.53  |  76.47  |  72.98  |
> | TypiClust  |  100  |  81.17  |  84.87  |  59.73  |  73.03  |  88.63  |  65.80  |  88.40  |  78.20  |  80.13  |  94.00  |  79.40 |
> | NTKCPL(self)  |  100  |  81.73  |  92.53  |  69.13  |  50.00  |  82.97  |  78.77  |  85.30  |  87.50  |  91.40  |  92.10  |  81.14 |
>
> **For question 1:**
>
> We appreciate your suggestion regarding the exploration of NLP datasets. In this current work, we focus on computer vision tasks. This decision was made to maintain consistency with previous research on low-budget active learning [R1].
>
> Nevertheless, we find your suggestion important and believe that investigating the performance of our approach on NLP data could be a valuable direction for future research. We will consider NLP datasets in our future work to provide a more comprehensive evaluation of our method's capabilities.
>
>
> **For limitation 1:**
>
> Thank you for pointing this out.  The current approximations primarily stem from the NTK (neural tangent kernel) approximation of DNNs and the utilization of subsets to estimate empirical risk over the entire active learning pool. As the reviewer pointed out, reducing these approximation errors, especially the latter one,  requires more computational resources.
>
> [R1] Hacohen, Guy, Avihu Dekel, and Daphna Weinshall. "Active Learning on a Budget: Opposite Strategies Suit High and Low Budgets." International Conference on Machine Learning. PMLR, 2022.

---

### Official Review · Reviewer_YYhz · 2023-07-05

**Soundness:** 2 fair
**Presentation:** 1 poor
**Contribution:** 2 fair
**Rating:** 4
**Confidence:** 3

**Summary:**

Active Learning is a crucial problem that focuses on selecting a subset of examples from an unlabeled dataset to be labeled. The primary objective is to ensure that when the model is trained using these selected examples, it achieves a lower empirical risk upon evaluation, assuming all the unlabeled data points are eventually labeled.

While getting labels is a difficult task, current foundation models that utilize self supervision are on-par with many supervised learning procedures. This work aims to use the existing self supervised trained models as the feature extractor, to train a classifier network with the active learning. To get the samples to train the neural network, the paper proposes to use NTK to get the classifier's output, if it was trained on an example say $x^{\prime}$ from the unlabeled pool. Then based on a criterion that depends on the accuracy ( 0-1 loss ), algorithm returns a set to be labeled.

This process is done iteratively. Paper shows improvement against common baselines such as Random/Coreset/BADGE/Entropy/Lookahead and shows that there are improvements. Overall, the problem is well motivated, however, there are several confusions and in particular writing issues, that makes this submission not fit at this stage to be published.






**Strengths:**

- Idea of using NTK on top of self supervised trained network is good, as it also provides a scope for theoretical guarantees.
- Wide spectrum of baseline has been covered, and gains are decent in low budge regime.
- Adaptive strategy for refining clusters is interesting.

**Weaknesses:**

Major weakness of this work is lack of clarity and writing. The notations at many places are not clear to me at all, with interchanging subscripts with "t" time and indexing for label at many places. I will summarize the places where there is clear notational abuse, or inconsistency in the upcoming lines. While the work is interesting, the mathematical notations and algorithm needs to be written very precisely and clearly, which in current form does not seem fit for the publication.

1. Section 2 never formally defines what $f$ is mathematically. What is the input space and what is the output space? Does it output class logits, or features?
2. Wherever an $\operatorname{argmin}$ is written, it should always have the space of optimization
3. Section 3.2 near equation 4, it is written "We denote the predictions of NTK with the dataset $D_{C}$ as $\hat{f}_{DC}$". then does $f$ also outputs a class, or a one hot vector?
4. In summary the proposed algorithm NTKCPL chooses examples to be labeled such that when added to the labeled pool, they'd minimize the empirical risk? Since the backbone is not trained, and they're also using NTK, how different is it in spirit from Mohamadi et al?
5. In NTKCPL Algorithm, I don't understand what exactly is $f_{self}$ and $f_{al}$. It is never defined. Is it the feature extractor and the learned classifier? If yes, then why define a new variable?
6. In NTKCPL Algorithm line 6, $b_i$ is never defined.
7. In NTKCPL Algorithm line 13, subroutine for NTK calling is never defined. Moreover, it seems like an overloading of $\hat{f}$ argument. Lastly, there doesn't seem to be utility of $f_{0}$ other than this routine.
8. Line 20 seem to be performing vacuous setminus from the unlabeled pool, where as in the labeled pool it seems incorrect to have a tuple of input and its pseudolabel as well.
9. Where is $f_t$ being used? There needs to be a full subroutine of AL procedure starting from scratch (that is $L = \emptyset$ going all the way to the required budget, in batches, if needed).
10. $g$ is never mathematically defined in the proposition.
11. I don't understand the meaning of dominant labels. Moreover, the usage of $D_{dom}$ is incorrect if it is indexing over the set of class labels, as previously $D_{.}$ is being used for the dataset.
12. Lots of unnecessary notations introduced such as $ymap$ which can be avoided by appropriately defining $g$
13. What is the meaning of $nff$ subscript, and similarly, $fnf$?
14. Why should clusters based on classifier being trained be reliable? That is usage of $f_{al}$?
15. Experiments did not mention the pretrained data for each of the dataset/arch, neither mention about the MLP arch.
16. What's the reason for Entropy and other popular methods to underperform even at decently high budget such as order of 1000s?
17. How is max number of clusters determined?
18. How different is coverage estimation from accuracy?

Lastly, the work would've been benefited, if there were experiments from CLIP models, which are one of the most popular available pre-trained models.

**Questions:**

Please refer to the weakness section.

**Limitations:**

I think the paper should've had the use-cases where experiments involve pre-trained CLIP ResNet models.

---

> ### Author Rebuttal · Authors · 2023-08-10
>
> Thank you for pointing out the confusing notation, we will ensure that the revised version incorporates these corrections.
>
> **1.** It outputs class logits. In paper, the $f$ denote a neural network model, $f: \mathbb{R}^{d}\rightarrow \mathbb{R}^{k}$, which maps a input sample $x \in \mathbb{R}^{d}$ to $k$-class prediction.
>
> **2.** Thank you for pointing this out. We will fix it in the revised version of paper.
>
> **3.**  $\hat{f}$ is the output of the NTK model, which also outputs a class prediction.
>
> **4.** Similar to the work of Mohamadi et al, our approach also falls within the category of "look-ahead" active learning methods based on the NTK. However, a key distinction lies in the sample selection criterion: while they employ an expected maximum change principle for sample selection, our method is rooted in a more fundamental objective – minimizing the expected risk of the entire active learning pool. Our primary technical contribution lies in the construction of CPL to realize this sample selection criterion. Building upon an error analysis of approximating empirical risk on the active learning pool using NTK and CPL, we develop an adaptive refining method for constructing CPL, which splits low-purity clusters and remains high-quality clusters. Additionally, our experimental results demonstrate a clear superiority of our approach over their work.
>
> **5.** $f_{self}$ is the output of the pre-trained self-supervised model's backbone, while $f_{al}$ corresponds to the output of the penultimate layer of the 2-layer MLP classifier (linear-bn-relu-linear) in our paper. Notably, $f_{al}$ denotes the output of the first linear layer within the 2-layer MLP classifier. To avoid any potential confusion between the neural network model and the feature representations, we intend to clarify our notation in the revised version of the paper.
>
> **6.** Thank you for pointing this out. It's a typo. It should be $b$, the budget of active learning.
>
> **7.** In line 13, we compute the output approximated by NTK directly following Equation (13) of the paper. The NTK kernel, $ker$, and the output of the neural network, $f_0$, at the initialization parameters are solely used for the computation of the output approximated by NTK.
>
> **8.** The sample selection is done in line 18, and it is the basic active learning loop: query the true label of selected samples from oracle (line 19), merge them into the existing labeled set (line 20), and train a new model based on the extended labeled set (line 21). In line 19, $ x_{i'_{1,...,b}} $ refers to the $b$ samples to be labeled selected by the active learning algorithm.
>
> $ y_{i'_{1,...,b}} $  is their true label from the oracle
>
> The CPL is write with subscript $cpl$ like $y_{cpl,i}$
>
> **9.**  Thank you for pointing this out. It's an overloading definition. The $f_t$ is the classifier training with the labeled set at the $t$ active learning cycle. We will revise the algorithm with the corrected notation.
>
> **10.**  We put the revised definition of $g$ in the global response pdf file.
>
> **11.** The term "dominant label" refers to the true class that has the most samples among the certain CPL class. We will fix it in the revised version of the paper.
>
> **12.** Thank you for pointing this out, we will fix it in the revised version.
>
> **13.** As line 177 of the paper, the probability that the NTK prediction agrees with the $y$ but not with $y_{cpl}$ as $P_{fnf}$, and the probability that the NTK prediction does not agree with $y$ but agrees with $y_{cpl}$ as $P_{nff}$. The first $f$ or $nf$ denotes whether NTK prediction agrees with the true label, $y$. And the last $f$ or $nf$ denotes whether NTK prediction agrees with the true label, $y_{cpl}$.
>
> **14.** As mentioned in our response to question 5, $f_{self}$ represents the output of the self-supervised pre-trained model's backbone, while $f_{al}$ corresponds to the output of the hidden layer of the 2-layer MLP classifier, where the classifier is trained on top of the frozen self-supervised pre-trained backbone.
>
> As discussed in the paper [R1], training an MLP classifier on the frozen pre-trained backbone often yields better performance compared to using a Linear classifier. This observation suggests that the MLP classifier learns features more suitable for classification. In other words, after being fine-tuned with some labeled data, the active learning feature $f_{al}$ may be more tailored for classification tasks than $f_{self}$. Consequently, clustering on the $f_{al}$ feature space is likely to result in higher-quality pseudo-labels, thereby enhancing the effectiveness of our active learning method.
>
> **15.** The Oxford-IIIT Pets and ImageNet-100 experiments employed a ResNet-50 model pre-trained on the ImageNet dataset. As for CIFAR-10, CIFAR-100, and SVHN, we performed pre-training on their respective datasets using the following architectures: ResNet-18 for CIFAR-10, WRN-28-8 for CIFAR-100, and ResNet-18 for SVHN. The architecture of the pre-trained models was written in the Implementation section of the paper.
>
> Regarding the architecture of the MLP classifier, it was written in Appendix Section 3, Table 1. The architecture consists of a 2-layer MLP with the following structure: Linear + BatchNorm + ReLU + Linear. The output dimension of the first linear layer is specified in Appendix table 1 too.
>
> **16.** The likely reason is the training method. In our study, we freeze the backbone and train an MLP classifier, which demonstrates superior performance within the low-budget regime. We add some extra experiments of fine-tuning the entire network. The results are shown in table 10 of the global response pdf file.
>
> **CLIP model**. Thank you for pointing this out. We will consider it in future work.
>
> [R1] Ren, Yi, et al. "How to prepare your task head for finetuning." The Eleventh International Conference on Learning Representations. 2022.

---

> > ### Author Response · Authors · 2023-08-12
> >
> > **17.** It's a hyperparameter. We roughly set the maximum number of clusters to be around 3-5 times the number of classes in the dataset. For datasets with a larger number of samples per class (such as CIFAR-10 and SVHN), we increase the maximum number of clusters to around 10 times the number of classes in the dataset.
> >
> > **18.** Coverage typically refers to a region identified by the active learning algorithm, within which the samples often exhibit higher accuracy. It does not align perfectly with the true accuracy of the active learning pool. We argue this discrepancy is a limitation in previous active learning methods, which is why we propose estimating accuracy on the active learning pool based on NTK and CPL. This approach aims to align coverage with the actual accuracy of the active learning pool, enhancing performance.

---

> > > ### Comment · Reviewer_YYhz · 2023-08-17
> > > **Thanks for the rebuttal.**
> > >
> > > I thank the authors for their thoughtful rebuttal, and I have thoroughly reviewed the paper again, taking into account the errata they provided (and responses to other reviewers). While I am inclined to raise my rating by a point, it is important to note that substantial revisions are still required for the paper to achieve a higher level of clarity and coherence. Consequently, I am unable to consider raising my rating beyond this point.

---

### Official Review · Reviewer_UZLm · 2023-07-05

**Soundness:** 3 good
**Presentation:** 2 fair
**Contribution:** 3 good
**Rating:** 6
**Confidence:** 2

**Summary:**

The paper presents a look-ahead strategy for more efficient active learning when used with self-supervised learning features. The approach uses Neural tangent kernels and pseudo-labels generated by clustering self-supervised or active learning features to estimate an approximation of the empirical risk of each unlabeled data sample. It then selects those examples for label annotation that will maximally reduce this empirical risk in each iteration. The paper demonstrates the validity and performance of this approach on 5 image datasets, showing that the approach outperforms other baseline active learning methods in most cases, and remains dominant over a larger range of training budgets over earlier SOTA methods for low and high budget strategies.


**Strengths:**

Originality: To my knowledge this is the first look ahead strategy for active learning that combines a neural tangent kernel with clustering based pseudo-labels to estimate an approximation of empirical risk of each unlabeled data sample to select for  active learning.

Quality: The paper is of average quality. It is incremental, building on earlier concepts of NTK and applying it to active learning with self-supervised features. It does a good job motivating, and validating the approach. The presentation in section 3.3 with a lot of new terms and notations is tiring and could be simplified.

Clarity: The paper could be improved with a few more editing passes to complete some incomplete sentences and polish the grammar for better readability.

Significance: The work is significant since it shows an approach that is at or improves on SOTA for active learning with self-supervised features.


**Weaknesses:**

I see the following weaknesses:

1. Section 3.3 is currently dense with a lot of new terms and notations that are not motivated or explained well. I suggest the authors refine this section, explaining more how the arguments lead to their proposition. The appendix is not very helpful as it currently stands to understand this proposition well.

2. A major dimension missing in the paper is how the time taken for active learning with NTKCPL compares with other SOTA methods in the different budget regimes. I suspect that NTKCPL is faster, but it is not clear if it indeed is faster, and if so, by how much. Since there is also a clustering algorithm run for each iteration, it is not clear how the overall approach scales with size and dimensionality of data, number of classes etc.

3. In Algorithm 1, I think line 6 should read “min(b_0/2,”. It now reads “min(b_i/2,”

4. Figure 34 2 compares various methods with NTKCPL. However, different panels use different colors for the same method, making it hard to read the (already dense) plots. Please use consistent colors/line types for the same method in each plot



**Questions:**


1. From the weaknesses above, how does NTKCPL scale with size and dimension of data, number of classes etc., and how do the running times compare with SOTA methods over different budget regimes?

2. Section 4- implementation: The paper mentions 2 sets of models used for each of the 5 datasets. What do the 2 sets represent - is one used for high budget and the other low budget?


**Limitations:**

I don’t see any significant negative societal impact of this work.

---

> ### Author Rebuttal · Authors · 2023-08-10
>
> Thank you for your constructive reviews, which will help us to improve the quality of the paper.
>
> **Weakness 1**
>
> We appreciate the reviewer's feedback regarding Section 3.3. We will carefully revise it to provide a more comprehensive and intuitive explanation of the arguments that lead to our proposition in the revised version.
>
> **Weakness 2**
>
> In general, as our approach falls under the category of "Look-ahead" active learning methods, the computational cost is generally higher compared to "myopic" type methods like entropy. However, when compared to other "Look-ahead" active learning methods, our approach demonstrates similar time complexity. The analysis of the time complexity is shown in the global response. And the practical running time of our method and one of the baseline methods, LookAhead, is shown in Appendix section 4.
>
> **Weakness 3**
>
> Thank you for pointing out the issue of typo. It's min($b/2$, where the $b$ is the budget of each active learning cycle. We will fix them in the revised version.
>
> **Weakness 4**
>
> Thank you for pointing out the issue of inconsistent figure color. We will fix them in the revised version.
>
> **Question 1**
>
> The analysis of the time complexity is shown in the global response.
>
> **Question 2**
>
> For the two different pretraining methods used in our study, we employed the BYOL for the larger dataset, ImageNet-100. BYOL demonstrated superior pretraining performance compared to SimSiam on the large dataset. However, the BYOL paper did not provide pretraining results for small-scale datasets, such as CIFAR-10. So, we choose Simsiam to pre-train on these small-scale datasets. Simsiam reported good pertaining performance on the small dataset.
>
> Regarding the selection of different network architectures for each dataset, we followed established practices from previous research. For datasets with lower image resolutions, such as CIFAR-10 and CIFAR-100, we utilized the ResNet18 or WRN288 architecture. For datasets with higher image resolutions and greater complexity, like ImageNet-100, we adopted the ResNet50 architecture.

---

> > ### Comment · Reviewer_UZLm · 2023-08-17
> >
> > Thanks for addressing my earlier review comments.
> > I will keep my earlier rating recommendation

---

### Official Review · Reviewer_d28k · 2023-07-06

**Soundness:** 3 good
**Presentation:** 3 good
**Contribution:** 3 good
**Rating:** 5
**Confidence:** 3

**Summary:**

The paper proposes a active learning strategy that combines self-supervised learning with NTK approximation to estimate empirical risk more accurately. The proposed method outperforms state-of-the-art methods and has a wider effective budget range.

**Strengths:**

Well-written: The paper is well-written, informative, and easy to understand. The authors provide clear explanations of the proposed method and the analysis, making it accessible to a wide audience.

Comprehensive analysis: The paper presents a comprehensive analysis of the proposed method, including an ablation study and experiments on various datasets.

Experimental results: The paper presents experimental results that demonstrate the effectiveness of the proposed method on various datasets. The results show that the proposed method outperforms state-of-the-art methods in most cases and has a wider effective budget range.


**Weaknesses:**

* Table 2 is misssing

* More self-supervised learning method + active learning should be compared

* Novelty seems not strong enough for NIPS, as the author mentions, self-supervised + active learning has been worked before, and would you explain why the previous methods are not competitive as this



**Questions:**

* “in our scenario, training on top of the self-supervised model, NTK does not approximate predictions of the whole network well. The main
 reason is that weights of the neural network are initialized by self-supervised learning rather than
NTK initialization, i.e., drawn i.i.d. from a standard Gaussian” ，I still confused about the reason, would you give me more detail?
* For Figure 1, Is it a concept graph or an experimental result graph?


**Limitations:**

The proposed method requires a pre-trained self-supervised model, which may not be available or feasible to obtain in some scenarios. This limits the applicability of the proposed method to certain domains and datasets.

Additionally, the paper does not provide a detailed analysis of the computational complexity of the proposed method, which may be a concern in some scenarios where computational resources are limited.

---

> ### Author Rebuttal · Authors · 2023-08-10
>
> Thanks so much for your constructive reviews.
>
> **1. For weakness (1): table 2 is missing**:
>
> Table 2 was included in line 273 of the paper.
>
> **2. For weakness (2): More baseline**:
>
> Thank you for your comment. We have reported the TypiClust that is tailored for this specific setting in the paper. To include more baseline, we add a new method, probcover, which is designed specifically for the low-budget regime. The results of the new baseline are shown in Tables 1, 2, 3 of the global response PDF file.
>
> **3. For weakness (3): Novelty seems not strong enough for NIPS, as the author mentions, self-supervised + active learning has been worked before, and would you explain why the previous methods are not competitive as this**:
>
> In terms of novelty, our method combines NTK and CPL to estimate the empirical risk of the active learning pool. And the approximation error is analyzed. So, unlike many prior active learning methods that rely on heuristic criteria, our method offers a certain degree of theoretical justification (as pointed out by reviewer YYhz).
>
> Moreover, the main drawback of existing AL+SSL methods is the narrow effective budget range. Many of these methods have been assessed in scenarios where the annotation budget is extremely limited (i.e., labeled samples are fewer than 6 times the number of dataset classes). In that case, the performance of the models is often not good enough. However, as the number of annotated samples gradually increased, we noted a diminishing performance of existing low-budget active learning strategies. In some instances, these strategies show lower performance than the random baseline. This situation often leads us to encounter a challenge when employing existing low-budget active learning strategies. These observations motivate us to propose a novel approach with a wider effective budget range within the low-budget regime.
>
> In Section 2, we visualize our insight, where we identify a key issue in existing methods rooted in feature distance-based coverage estimation. This can lead to either overestimation or underestimation, resulting in suboptimal sample selection and limiting the effective budget range. To address this, we propose using the NTK and CPL for estimating the empirical risk on the active learning pool.
>
> **4. For question (1): The reason why NTK does not approximate a NN initialized by pre-trained weights very well**:
>
> Existing methods approximate the neural network model output using the NTK by performing a first-order Taylor expansion around the network's initial values [R1] (sec. 2.2). NTK theory posits that as the width of a neural network tends to infinity, the changes in each weight during training approach zero, making the Taylor expansion around the initial values sufficiently accurate. However, in practical scenarios, the practical neural networks have finite widths, and the changes in weights during training cannot be regarded as zero. Thus the choice of the Taylor expansion point impacts the final approximation error.
>
> In our training setting, we initialize the neural network with weights from self-supervised pretraining. After fine-tuning the entire network with labeled data, the weights of the network are significantly different from the random initialization weights used by NTK. As a result, NTK cannot provide a satisfactory approximation of the output of neural networks trained in this manner.
>
> **5. For question (2): For Figure 1, Is it a concept graph or an experimental result graph?**:
>
> In fig.1, all unlabeled samples are visualized using t-SNE based on CIFAR-10 self-supervised features. The labeled samples consist of 50 samples selected by probcover [R2]. In fig.1(a), the radius of blue circles is computed using the coreset approach. In fig.1(b), the coverage radius is calculated using the probcover method. In fig.1(c), the black dots represent samples deemed covered by our method, i.e., samples for which NTK predictions align with CPL. In fig.1(d), the black dots represent true covered samples that are predicted consistently with the true labels by a classifier trained using this set of labeled samples.
>
> **6. For limitation (1): requires a pre-trained self-supervised model**:
>
> To our knowledge, a major constraint on obtaining self-supervised models lies in the requirement for a sufficient amount of unlabeled data to train them. However, there are scenarios where obtaining an adequate volume of data may be challenging. In response, our experiments encompass the utilization of self-supervised models pre-trained on ImageNet for the Oxford-Pets dataset, yielding promising results as illustrated in Fig. 2(d) of the paper.
>
> Moreover, while our experiments are conducted based on self-supervised models, our proposed method is not confined solely to self-supervised models. It is applicable in contexts where well-pretrained models are available. Given the extensive research and rapid advancement of foundational models, we believe that the constraints associated with obtaining a high-quality pre-trained model are progressively diminishing.
>
> **7. For limitation (2): computational complexity**:
>
> Thank you for pointing this out, we analyze it in the global response.
>
> [R1] Lee, Jaehoon, et al. "Wide neural networks of any depth evolve as linear models under gradient descent." Advances in neural information processing systems 32 (2019).
>
> [R2] Yehuda, Ofer, et al. "Active learning through a covering lens." Advances in Neural Information Processing Systems 35 (2022): 22354-22367.

---

> > ### Comment · Reviewer_d28k · 2023-08-16
> > **Reply to authors**
> >
> > Thanks for your response. I read the paper once agin and all the review comments and replies. I agree with Reviewer YYhz that the mathematical notations should be made clearer. Have you submitted a new version?

---

> > > ### Comment · Area_Chair_3msY · 2023-08-16
> > >
> > > Reviewer d28k, just to clarify, there is no "revision" process for this track. So we need to make a decision based on the current information (and our estimate on the possible improvement during camera-ready). Thanks.
> > >
> > > Your AC

---

> > > ### Author Response · Authors · 2023-08-17
> > > **Notations**
> > >
> > > Thank you for your response. The revised notations are shown in the global response. We will incorporate them into a revised version of the paper.

---

> > > > ### Comment · Reviewer_d28k · 2023-08-17
> > > > **Reply to author**
> > > >
> > > > Thanks for your reply and the clearer notations, and I would like to increase the score.

---

### Official Review · Reviewer_Pudn · 2023-07-08

**Soundness:** 3 good
**Presentation:** 2 fair
**Contribution:** 4 excellent
**Rating:** 6
**Confidence:** 3

**Summary:**

This paper introduces a novel approach that combines active learning with self-supervised learning, known as neural tangent kernel clustering-pseudo-labels (NTKCPL). The method leverages the power of the neural tangent kernel (NTK) in conjunction with self-supervised learning features to enhance the estimation of lookahead. Additionally, clustering-pseudo-labels are employed to estimate the classification error. The paper includes a thorough analysis of the approximation and presents comprehensive experimental results, comparing the proposed methods against benchmark techniques.

**Strengths:**

As demonstrated in the comparison experiments, NTKCPL exhibits substantial performance gains.

The analysis of CPL error is important, as it effectively reveals that errors arise from over-clustering and impurity. Building upon this analysis, the proposed cluster generation algorithm effectively addresses these issues, displaying a logically concrete solution.

**Weaknesses:**

The paper is motivated from the concepts of "phase transition" and "effective budget range" in active learning. However, it lacks analysis regarding why the proposed method can increase the effective budget range. Additionally, in Table 2, the absolute percentage of the "Effective Budget Ratio" is dependent on the chosen total annotation quantity, e.g. a larger total annotation quantity leads to a smaller "Effective Budget Ratio," suggesting that the "Effective Budget Ratio" is not well-defined.

**Questions:**

Eq. 6 may include a typo: Shouldn't the superscript "k" be a subscript "k"? And j is not shown in the right hand side of the equation.

What's the difference between $\hat{f}_{y}$ and $\hat{f}_{ymap}$



**Limitations:**

The authors have addressed the limitations of this paper.

---

> ### Author Rebuttal · Authors · 2023-08-10
>
> Thank you for your valuable reviews.
>
> **1. Weakness (1) “lacks analysis regarding why the proposed method can increase the effective budget range”:**
>
> In Section 2, we visualize our insight, where we identify a key issue in existing methods rooted in feature distance-based coverage estimation. This can lead to either overestimation or underestimation, resulting in suboptimal sample selection and limiting the effective budget range. To address this, we propose using the NTK and CPL for estimating the empirical risk on the active learning pool. Our results in Fig. 4 showcase that our method's estimated empirical risk closely approximates the true empirical risk. This alignment likely contributes to the extension of the effective budget range facilitated by our approach.
>
> Moreover, further comprehensive investigations remain a fascinating direction for designing robust and safe active learning strategies.
>
> **2. Weakness (2) “Effective Budget Ratio" is not well-defined”:**
>
> Indeed, the current definition of effective budget ratio may not be an ideal metric, as it can be influenced by the total number of annotations. Exploring better ways to quantify the effective annotation range for active learning strategies could be a valuable direction for future research. Nevertheless, as a qualitative indicator, this metric can still serve as a valuable tool. Regardless of the total annotation count, if an active learning strategy exhibits a notably low effective budget ratio, it suggests that the strategy might not reliably yield positive outcomes. Hence, the reported results in this paper still demonstrate the advantages of our proposed method over typical existing active learning strategies.
>
> Furthermore, the effective budget ratio reported in the paper serves as a succinct summary of all experiments. The detailed effective budget scope can be found in Fig. 3.
>
> **3. Question eq.6**
>
> Thank you for pointing this out. We clarify the definition of eq.6 in the global response pdf file.
>
> **4. Difference between $\hat{f}{y}$ and $\hat{f}{ymap}$**
>
> The $\hat{f}{y}$ is the output of NTK trained with the true label and $\hat{f}{ymap}$ is the output of the label mapping function $g$, which maps the output of the NTK trained with the CPL label into the true label class.

---

### Official Review · Reviewer_G9yf · 2023-07-26

**Soundness:** 3 good
**Presentation:** 2 fair
**Contribution:** 3 good
**Rating:** 5
**Confidence:** 3

**Summary:**

This paper aim to develop an active learning method effective across various budgets and compatible with self-supervised learning. The proposed approach, NTKCPL, a look-ahead active learning strategy, selects a subset that are expected to train network to minimize error of unlabeled data pool. For efficiently estimate the model prediction trained with candidate set, they employ NTK. To do so, they freeze the network's backbone and train only the classifier. Pseudo labels are assigned to the unlabeled data pool for empirical risk calculation, achieved by applying a constrained K-means algorithm to self-supervised features. The loss between pseudo labels and approximated predictions is calculated to select data that will likely helpful to minimize the unlabeled pool's loss. The proposed method is evaluated on five datasets and outperformed the baselines on most of the datasets and budget ranges.

**Strengths:**

- Combining NTK and pseudo label from clustering to estimate empirical risk is new to me.
- The proposed method is evaluated on various datasets, and in most cases, it demonstrated superior performance compared to the baseline approaches.
- They analyze the approximation error of the empirical risk when using NTK and CPL

**Weaknesses:**

- Limited technical contribution
    - An essential element of the proposed method stems from earlier work [26] that utilizes NTK approximation of DNN prediction for look-ahead active learning.
    - The most prominent distinction from [26] is that this work utilize expected error reduction instead of expected model output change for active selection, and proposed a method for assigning pseudo labels to facilitate this.
    - Given that both expected error reduction and pseudo labeling through feature vector clustering are widely used techniques, the technical contribution of the proposed method could be seen as incremental.
- Concerns about practicality
    - The method needs to freeze the backbone to ensure the accuracy of the NTK approximation (line 151 - 154). However, according to the existing self-supervised learning literature [a], there is a substantial performance gap between fine-tuning the entire network and those that only train the classifier.
    - Moreover, the proposed method appears to be dependent on the quality of the features learned through self-supervised learning. Although it is claimed that the active learning feature is used to improve clustering purity, the results in table 1 reveals a performance advantage for self-supervised features in low-budget situations.
- Need for comprehensive baseline comparisons
    - It would be beneficial if the performance of [24] was also reported, given that the proposed method follows the training configuration of [24].
    - It appears that the results using self-supervised features on Cifar 100 may be missing. It would be advantageous to see these results.
    - Furthermore, I am curious as to whether the baseline methods were also trained with the frozen backbone, and how their performance would vary depending on this factor.
- I encountered difficulties in smoothly following the provided script. I believe there is room for improvement in the writing.
    - I found the paper's main point to be somewhat confusing. On lines 49 - 52, one of the stated goals of the proposed method is an active learning strategy with a wider effective budget range, yet on line 226, there is a focus on the low-budget regime.
    - Furthermore, the notations in the method section have not been clearly defined, and they are somehow confusing. For instance, in the script, 'f' denotes a neural network (line 125), 'f_0' represents the network's output (line 127), 'f_self' and 'f_al' are features (line 1 of algorithm 1), 'f_t' signifies the classifier (line 21 of algorithm 1), and '\hat(f)_cpl' is the prediction (line 170).

[a] He, Kaiming, et al. "Masked autoencoders are scalable vision learners." *Proceedings of the IEEE/CVF conference on computer vision and pattern recognition*. 2022.

**Questions:**

- I am interested in understanding the performance degradation when not using a frozen backbone.
- I am curious about the upper bound of performance achieved through empirical risk estimation using true labels and NTK.

**Limitations:**

The paper appropriately states its limitations and broader impacts.

---

> ### Author Rebuttal · Authors · 2023-08-10
>
> Thank you for your valuable reviews, which helps us to improve the paper.
>
> **1. Weaknesses. technical contribution**
>
> Yes, our approach is built on the LookAhead framework. However, the key innovation is that we use the CPL to estimate the empirical risk and design a CPL construction method based on the analysis of the approximation error in estimating the empirical risk using NTK and CPL. Table 1 and Figure 2 in the paper show that our method significantly outperforms LookAhead.
>
> While it is true that expected error reduction (EER) is an established concept, its application within the context of deep active learning remains relatively underexplored. To the best of our knowledge, we have only identified one previous work [R1] that incorporates EER in deep active learning. Furthermore, it's important to note that [R1] requires an extra validation set to estimate expected error, which may not be practical in the low-budget regime. In contrast, our proposed method leverages CPL to estimate expected error.
>
> Additionally, regarding technical implementation, our CPL is different from a simple execution of clustering in the feature space. We carefully analyze the approximation error and design an adaptive refining method for CPL. This approach reduces the impact of these approximation errors and enhances the accuracy of our method.
>
> **2. Weaknesses. Concerns about practicality (a) training method**
>
> Fine-tuning the entire network yields significantly better performance when using a large amount of annotated data. However, we would like to emphasize that our main focus in this paper is on the low-budget regime with a wider effective budget range explored in the previous paper [R2].  Within this range of annotation quantities, training only the classifier has demonstrated either superior or comparable performance compared to fine-tuning the entire network. The results of two training methods on CIFAR-10, CIFAR-100 and ImageNet-100 are shown in table 7, 8, 9 of the global response PDF file.
>
> **3. Weaknesses. Concerns about practicality (b) dependent on the quality of the self-sup. features**
>
> In the context of the five datasets used for validation, CIFAR-10 stands out as the only example where NTKCPL(self), our method based on self-supervised feature, significantly outperforms NTKCPL(al), our method based on active learning feature, in low-budget situations. We believe this difference in performance is attributed to the width of the active learning features (output of hidden layer of the 2-layer MLP classifier). In our CIFAR-10 experiments, we followed the approach of previous works, employing a 2-layer MLP classifier with a hidden layer width of 64, i.e. active learning feature dimension is 64, which is much smaller than the 512-dimensional self-supervised features.
>
> We conducted additional experiments on CIFAR-10 by adjusting the MLP hidden layer width to 512. After the number of annotations is greater than 100, the accuracy difference between NTKCPL(self) and NTKCPL(al) is less than about 0.5\%. (Due to space limitations, we will post the results during discussion)
>
> **4. Weaknesses. Comprehensive baseline (a)**
>
> When attempting to reproduce the method of [24], we encountered similar difficulties as described in [R2] (Appendix. D.4). Due to the unavailability of source code, reproducing the method reliably became challenging. To include more baseline methods, the results of probcover [R2] are shown in table 1, 2, 3 of the global response PDF file. As mentioned in [R2], its performance surpasses that of W-dist[24].
>
> **5. Weaknesses. Comprehensive baseline (b)**
>
> Thank you for pointing this out.  The result is shown in table 4 of the global response PDF file. We will include it in the revised version of paper.
>
> **6. Weaknesses. Comprehensive baseline (c) and Question 1**
>
> Yes, the baseline methods were implemented with the frozen backbone too. We observed that baseline performance of training only the classifier outperforms that of fine-tuning the entire network in the low-budget regime (about $20-50$ times the number of label classes in the dataset).
>
> We add the baseline performance with fine-tuning shown in table 10 of the global response PDF file. Because for the ImageNet-100 dataset, training only the classifier consistently achieves higher accuracy compared to fine-tuning for random baseline within our annotation quantities range. We add experiments on CIFAR-10. When the fine-tuning is adopted, the interval where our method outperforms other baseline methods is roughly consistent with the interval where training only the classifier surpasses fine-tuning. Beyond this range, our method shows diminished performance compared to high-budget active learning baselines.
>
> **7. Weaknesses. Room for improvement in the writing (a)**
>
> In our work, we focus on the low-budget regime, and the mentioned "wider effective budget range" refers to the broader scope within this low-budget regime. In previous papers [R2], the typical low-budget regime usually refers that the total number of labels is about 6 times the number of classes in the dataset. However, the model's performance is often not satisfactory within this range. When increasing the number of labeled samples, existing active learning strategies do not reliably outperform the random baseline. Therefore, it became necessary to propose a new method that could achieve a wider effective budget range within the low-budget regime.
>
> **8. Weaknesses. Room for improvement of writing (b)**
>
> Thank you for pointing out the issue of confusing notations.  We will ensure that the revised version incorporates these corrections.
>
> **9. Question 2. Upper bound of performance**
>
> We have reported the accuracy achieved by directly substituting the CPL in our method with true labels in table 5, 6 of the global response PDF file. As the number of annotated samples increases, we observe that our method's performance approaching that of the utilization of true labels.

---

> > ### Author Response · Authors · 2023-08-12
> >
> > **9. Question 2. Upper bound of performance**
> >
> > Furthermore, it is important to note that even with the direct substitution of CPL with true labels in our method, we have not yet reached the performance upper bound of active learning using NTK to estimate empirical risk. This discrepancy can be attributed to two main factors. Firstly, we use a subset of the active learning pool to estimate empirical risk. Secondly, our current sample selection strategy still follows a greedy approach. Thus, we believe the potential for substantial performance gains if improvements are made in these two aspects for future work.
> >
> > [R1] Mussmann, Stephen, et al. "Active Learning with Expected Error Reduction." arXiv preprint arXiv:2211.09283 (2022).
> >
> > [R2] Yehuda, Ofer, et al. "Active learning through a covering lens." Advances in Neural Information Processing Systems 35 (2022): 22354-22367.
> >
> > **Table for 3. Weaknesses. Concerns about practicality (b)**
> >
> > **Table Accuracy of NTKCPL(self)  and NTKCPL(al) on CIFAR-10 when the classifier is the 2-layer MLP with hidden layer width 512. All results are averages over 3 runs.**
> >
> > | #Label | 20 | 40 | 60 | 80 | 100 | 200 | 300 | 400 | 500 | 1000 | 1500 | 2000 |
> > | ----------- | ----------- | ----------- | ----------- | ----------- | ----------- | ----------- | ----------- | ----------- | ----------- | ----------- | ----------- | ----------- |
> > | NTKCPL(self) | 48.84$\pm$2.99 | 66.23$\pm$1.54 | 74.04$\pm$1.20 | 78.97$\pm$1.23 | 79.40$\pm$0.67 | 83.14$\pm$0.52 | 84.62$\pm$0.33 | 84.76$\pm$0.22 | 85.10$\pm$0.72 | 87.09$\pm$0.32 | 87.62$\pm$0.24 | 87.89$\pm$0.06 |
> > | NTKCPL(al) | 49.67$\pm$2.93 | 64.24$\pm$2.90 | 72.10$\pm$2.38 | 76.67$\pm$0.64 | 78.78$\pm$0.92 | 82.66$\pm$0.81 | 84.13$\pm$0.44 | 84.67$\pm$0.23 | 85.53$\pm$0.20 | 87.32$\pm$0.33 | 87.75$\pm$0.19 | 88.13$\pm$0.32 |

---

> > > ### Comment · Reviewer_G9yf · 2023-08-14
> > > **Additional comments and questions**
> > >
> > > Thank you for the extensive responses to my questions and for conducting additional experiments. The experiment has addressed most of my concerns related to the experiment. However, I still have concerns regarding the technical contribution. The details are outlined below, and I request further clarification on these issues.
> > >
> > > ---
> > > I still find it challenging to agree with the CPL’s contribution based on the approximation error. My difficulty in understanding Section 3.3, especially the approximation error and the CPL method, might be a reason. For a better understanding of this contribution, I ask the following clarification questions:
> > >
> > > - What is the definition of \hat{f}\_{ymap}? Both in the main text and in the response about 'Pudn', it's mentioned as the result changed to the true label class. Is \hat{f}\_{ymap} the true label and does the mapping function ‘g’ transform \hat{f}\_{cpl} to \hat{f}\_y? Looking at Line 172 of the main paper, it's based on equation (6). However, it seems unlikely that equation (6) always returns the true label.
> > > - Even with the added definition of ‘g’ in the document of global response, my doubts aren't fully cleared. On observing equation (1) in the document of the global response, through ‘g’, the k-th prediction of \hat{f}\_{cpl}(x\_i) becomes \hat{f}\_{ymap}(x\_i).  Does this mean the input and output dimensions of this ‘g’ function differ? This seems inconsistent with equation (2) in the document on global response. Based on this equation, to obtain \hat{f}^j\_{ymap}(x\_i) not only the k-th prediction of \hat{f}\_{cpl}(x\_i) is required but all predictions, contrasting the definition in equation (1) in the document of global response.
> > > - Moreover, I'd like a more detailed explanation regarding the connection between the proposition conclusion in Section 3.3 and the decomposition of error\_{CPL}.
> > > - Additionally, why does increasing the number of clusters elevate the over-clustering error? (Line 189)
> > >
> > > The method claims to be grounded on a careful analysis of the approximation error for the CPL method. However, its rationale doesn't entirely convince me.
> > > - To address the under coverage, the proposed approach seems to adjust the cluster count heuristically to half the budget count.
> > > - Is the core contribution of the proposed method dividing the budget in half, using one half for clustering the entire sample set, and the other half for iteratively clustering clusters with low model prediction purity?

---

> > > > ### Author Response · Authors · 2023-08-15
> > > > **Definition of \hat{f}_{ymap}**
> > > >
> > > > The definition of $\hat{f}_{ymap}(x_i)$ is as presented in the global response PDF file, where eq(1) should be corrected to $\hat{f}\_{ymap}(x_i) = g(\hat{f}\_{cpl}(x_i))$. It does not represent the true labels. It is the output of the mapping function “g”. This mapping takes into account the distribution of true labels within each CPL cluster and transforms the predictions, $\hat{f}\_{cpl}(x_i)$, into the label space of true labels, which helps us to analyze the $error\_{CPL}$. The input of the function “g”, $\hat{f}\_{cpl}(x_i)$, is a $K\_{cpl}$-dimensional vector, where $K\_{cpl}$ denotes the number of CPL categories. The output is a $K\_{y}$-dimensional vector, where $K\_{y}$ represents the number of true label categories.
> > > >
> > > > To illustrate the function of the $g$, consider the following example. Let the true labels for $x_1,...,x_8$ be 0, 0, 0, 1, 1, 1, 1, and 1, respectively. The CPL labels are 0, 0, 1, 1, 1, 2, 2, and 2, respectively. CPL cluster 0 contains samples $x_1$ and $x_2$, both having a true label of 0. Therefore, the dominant class of CPL cluster 0 is 0. CPL cluster 1 includes samples $x_3$, $x_4$, and $x_5$, with true labels 0, 1, and 1, respectively. The most frequent true label within this cluster is 1, making the dominant class of CPL cluster 1 equal to 1. Similarly, the dominant class of CPL cluster 2 is 1.
> > > >
> > > > Once the dominant class is established, we can transform the CPL predictions to the label space of true label using equations (1) to (3) in the global response pdf file. For instance, when $\hat{f}{cpl}(x_i) = [1,0,0]$, $\hat{f}{ymap}(x_i) = g(\hat{f}{cpl}(x_i)) = [1, 0]$. When $\hat{f}{cpl}(x_i) = [0,1,0]$, $\hat{f}{ymap}(x_i) = g(\hat{f}{cpl}(x_i)) = [0, 1]$. When $\hat{f}{cpl}(x_i) = [0,0,1]$, $\hat{f}{ymap}(x_i) = g(\hat{f}{cpl}(x_i)) = [0, 1]$. When $\hat{f}{cpl}(x_i) = [0.1,0.3,0.6]$, $\hat{f}{ymap}(x_i) = g(\hat{f}{cpl}(x_i)) = [0.1, 0.9]$.

---

> > > > ### Author Response · Authors · 2023-08-15
> > > > **Connection between the proposition conclusion in Section 3.3 and the decomposition of $error_{CPL}$**
> > > >
> > > > As shown in equation (5) and line 166 of the paper, the $error_{CPL}$ is defined as $error_{CPL} = \dfrac{1}{N} \sum_{i \in D} | Loss(\hat{f}\_{y}(x_i),y_{i}) - Loss(\hat{f}\_{cpl}(x_i),y_{cpl,i}) |$. When employing the 0-1 loss (as shown in line 158 of the paper), there are four cases:
> > > >
> > > > (1) the $Loss(\hat{f}\_{cpl}(x_i),y_{cpl,i})  = 0$ and $Loss(\hat{f}\_{y}(x_i),y_{i})  = 1$;
> > > >
> > > > (2) the $Loss(\hat{f}\_{cpl}(x_i),y_{cpl,i})  = 1$ and $Loss(\hat{f}\_y(x_i),y_{i})  = 0$,
> > > >
> > > > (3) the $Loss(\hat{f}\_{cpl}(x_i),y_{cpl,i})  = 0$ and $Loss(\hat{f}\_y(x_i),y_{i})  = 0$,
> > > >
> > > > (4) the $Loss(\hat{f}\_{cpl}(x_i),y_{cpl,i})  = 1$ and $Loss(\hat{f}\_y(x_i),y_{i})  = 1$.
> > > >
> > > > Both case (3) and case (4) don't contribute to $error_{CPL}$, so we only consider case (1) and case (2). Thus, the $error_{CPL} = P_{nff} + P_{fnf}$, where the $P_{nff}$ denotes the probability that the NTK prediction does not agree with $y$ but agrees with $y_{cpl}$ (case (1)), the $P_{fnf}$ denotes the probability that the NTK prediction agrees with the $y$ but not with $y_{cpl}$ (case (2)).
> > > >
> > > >
> > > > It's worth noting that all the analyses presented below are conducted under the prerequisite of the proposition: that is, the true labels of labeled samples are the dominant classes in their corresponding CPL clusters.
> > > >
> > > > For case (1), $P_{nff}$, when the NTK model with CPL labels predicts $x_i$ as class $y_{cpl,i}$, i.e., $argmax$ $\hat{f}{cpl}(x_i) = y_{cpl,i}$, according to the proposition, the NTK model using true labels will predict $x_i$ as the corresponding dominant class of $y_{cpl,i}$, which is $argmax \hat{f}{y}(x_i) = $ $argmax$ $g(onehot( argmax \hat{f}{cpl}(x_i)))$. However, as the definition of the case (1), $y_i \neq argmax \hat{f}{y}(x_i)$, it indicates that the true label $y_i$ of $x_i$ is not the corresponding dominant class of $y_{cpl,i}$. This implies that the samples within this CPL cluster include at least two different true labels, namely $y_i$ and the corresponding dominant class of $y_{cpl,i}$. In other words, this CPL cluster is impure.
> > > >
> > > > For case (2), $P_{fnf}$, similarly, according to the proposition, using the NTK model with the true labels would predict $x_i$ as the corresponding dominant class of $y_{cpl,i}$. This implies that the class of true label $y_i$ contains the CPL cluster $argmax \hat{f}{cpl}(x_i)$. According to the definition of case (2), it's evident that the true label class $y_i$ also encompasses the CPL cluster $y\_{cpl,i}$. Based on the definition of case (2),  $y\_{cpl,i} \neq argmax \hat{f}\_{cpl}(x_i)$. This signifies that there are at least two different CPL clusters, $y\_{cpl,i}$ and $argmax \hat{f}\_{cpl}(x_i)$, present within this true label class. In other words, this true class has been over-clustered.
> > > >
> > > > Taking all the above analysis into consideration, the ideal characteristics of a CPL can be summarized by two points:
> > > >
> > > > (1) Each CPL cluster should be as pure as possible, meaning that each CPL cluster would ideally contain only one true label class, in order to minimize impurity error $P_{nff}$.
> > > >
> > > > (2) For each true label class, it's desirable to minimize the occurrence of being clustered into different CPL clusters, thereby reducing the risk of over-clustering error $P_{fnf}$.

---

> > > > ### Author Response · Authors · 2023-08-15
> > > > **Why does increasing the number of clusters elevate the over-clustering error?**
> > > >
> > > > As mentioned in the previous response, the over-clustering error, $P_{fnf}$, happens when any true label category contains multiple CPL clusters. Therefore, increasing the number of CPL clusters may make $P_{fnf}$ more likely. We can also show some empirical evidence if desired.

---

> > > > ### Author Response · Authors · 2023-08-15
> > > > **Some question about “the method claims to be grounded on a careful analysis of the approximation error for the CPL method”**
> > > >
> > > > Thank you for your response. Below are further clarifications regarding these issues. In order to facilitate your access to our replies for each question, we have used different threads to respond to different issues.
> > > >
> > > > ---
> > > >
> > > > In general, based on the findings from our analysis of the approximation errors introduced by using the CPL, we argue the ideal CPL should aim for each cluster to be as pure as possible, ideally containing only a single true label category. Additionally, for each true label category, minimizing the occurrence of being clustered into different CPL clusters helps to reduce the risk of over-clustering error， $P\_{fnf}$. In light of this, we designed a CPL generation approach that identifies, splits and re-clustering low-purity CPL clusters (thus reducing impurity error, $P_{nff}$) while preserving high-purity clusters within CPL (reducing the risk of over-clustering error, $P_{fnf}$). This approach stands in contrast to simply increasing the cluster count (by increasing k in k-means clustering) to improve cluster purity. The straightforward approach of increasing the cluster count may make high-purity clusters become multiple smaller clusters, which could lead to over-clustering error, $P_{fnf}$.
> > > >
> > > > **“adjust the cluster count heuristically to half the budget count.”**
> > > >
> > > > Yes, as you pointed out, we have employed a heuristic approach to set the hyperparameter, the maximum number of CPL clusters. This value is set as the minimum between half of the budget count and a manually pre-defined maximum cluster count (as described in Algorithm 1, line 7 of the paper). However, it's important to note that the essence of the CPL method is derived from the analysis of CPL approximation errors. The method aims to refine the low-purity clusters in the initial clusters while retaining high-purity clusters, rather than simply increasing the cluster count to improve purity.
> > > >
> > > > **“Is the core contribution of the proposed method dividing the budget in half, using one half for clustering the entire sample set, and the other half for iteratively clustering clusters with low model prediction purity?”**
> > > >
> > > > No, “dividing the budget into half, using one half for clustering the entire sample set, and the other half for iteratively clustering clusters” is not our proposed method. Based on the above responses, the consideration of the ideal CPL, we designed an iterative refine clustering method to generate CPL as outlined in Algorithm 2 of the paper. At each round of active learning, only one CPL is used. The generation of the CPL begins with constrained k-means to produce the initial clusters (total $C_0$ clusters), followed by iterations to identify and split impure clusters into two by re-clustering to improve their purity. And those clusters with high purity remain unchanged.

---

> > > > > ### Comment · Reviewer_G9yf · 2023-08-21
> > > > >
> > > > > Thank you for providing a comprehensive clarification. Through the rebuttal, the authors have addressed some of my concerns regarding the technical contribution of CPL. Many reviewers, including myself, believe the paper's technical writing needs improvement. However, I think the authors could clarify these concerns in their revised version using the responses given during the rebuttal. Consequently, I'm inclined to raise my score. I've taken into account both the comments from other reviewers and the responses given by the authors.

---

### Author Rebuttal · Authors · 2023-08-10

Thanks to all the reviewers for their valuable feedback. The computational complexity of our algorithm is a common issue for several reviewers, so we put the computational complexity analysis in the global response.

The computational complexity of our algorithm can be broken down into two main components. The first component is the generation of CPL as described in Algorithm 2 of the paper and the second component involves the utilization of NTK approximations. Let's denote the size of the labeled dataset as $L$, the size of the unlabeled dataset as $U$, the budget for selecting labeled samples per active learning round as $b$, and the number of parameters in the model as $P$ (in our case, the parameters of the MLP classifier used in NTK computation).

Referring to [R1], we can write that the complexity of computing the NTK kernel is $O(LUP + L^2P + L^3)$. The time complexity for selecting single samples from $U$ using the NTK is $O(UL^2)$. Consequently, the time complexity for computing the NTK kernel and selecting $b$ samples is approximately $O(b(LUP + L^2P + L^3 + UL^2))$.

Regarding the CPL computation, the primary computational complexity arises from the execution of k-means clustering $(C_{max} - C_0)$ times, where we split the most impure cluster into two, resulting in $k = 2$. Here, $C_{max}$ represents the maximum number of clusters, $C_0$ is the initial number of clusters, the maximum number of iterations for k-means is denoted as $I$, and the feature dimension used for clustering is represented by $d$. As shown in Algorithm 1, each active learning round involves generating CPL once and the selection of $b$ samples, resulting in an overall complexity of $O(b(LUP + L^2P + L^3 + UL^2) + UdI)$. In our scenario, the dominant term is $O(bLUP)$, which is similar to the LookAhead framework [R1], the practical running time is shown in appendix section 4.

[R1] Mohamadi, Mohamad Amin, Wonho Bae, and Danica J. Sutherland. "Making look-ahead active learning strategies feasible with neural tangent kernels." Advances in Neural Information Processing Systems 35 (2022): 12542-12553.

---

> ### Author Response · Authors · 2023-08-17
> **Revised Notations**
>
> Thank you to all the reviewers for their valuable feedback. In response to the reviewers' concern regarding notations, we have provided an updated version of notation along with corresponding equations. We intend to incorporate these clarifications into the revised version.
>
> ---
> **Data**
>
> $x$ denotes the input sample, $y$ denotes the true label for $K_y$ classes, and $y_{cpl}$ denotes the CPL label for $K_{cpl}$ classes. $U$ denotes the unlabeled dataset, and $L$ denotes the labeled dataset, $D={(x_{i},y_{i})_{i=1}^{N}}$ denotes the whole dataset with $N$ samples.
>
> **Model**
>
> The neural network model is denoted as $f_{\theta_{f}}(\phi_{\theta_{\phi}}(x))$, where $\phi_{\theta_{\phi}}$ represents the feature extractor initialized with pre-trained weights $\theta_{\phi}$, $f_{\theta_{f}}$ refers to the classification head with weight parameters of $\theta_{f}$. The entire neural network maps input samples $x \in \mathbb{R}^d$ to classification prediction $ y_{pre} \in \mathbb{R}^{K_{y}}$, where $K_{y}$ is the number of classes.
>
> **NTK model**
>
> The model that approximates $f_{\theta_f}$ using NTK (Neural Tangent Kernel) is denoted $\hat{f}$. In this paper, we denote the NTK model using true labels as $\hat{f}\_{y}$, and the NTK model using CPL labels as $\hat{f}\_{cpl}$. $\hat{f}\_{y}$ maps the input $\phi\_{\theta_{\phi}}(x)$ to the $K_y$ classes classification prediction $\hat{y}\_{pre} \in \mathbb{R}^{K_y}$, similar to $f\_{\theta_f}$. $\hat{f}\_{cpl}$ maps the input $\phi\_{\theta_{\phi}}(x)$ to the $K_{cpl}$ classes classification prediction $\hat{y}\_{pre, cpl} \in \mathbb{R}^{K_{cpl}}$.
>
> **Feature**
>
> The self-supervised feature $feas_{self}$ is the output of the pre-trained feature extractor $\phi_{\theta_{\phi}}(x)$, $feas_{self} = \phi_{\theta_{\phi}}(x)$. In this paper, $f_{\theta_{f}}$ refers to an MLP classifier, specifically. We denote the output of the penultimate layer of $f_{\theta_{f}}$ as the active learning feature $feas_{al}$.
>
> **Equation (1)**
>
> $ min_{D\_{C} \in  D} \dfrac{1}{N} \sum_{ (x, y) \in D} Loss(f_{\theta^*}(\phi_{\theta_{\phi}}(x)),y)$
>
> $s.t.  \theta^* \in argmin_{\theta} \dfrac{1}{N} \sum_{ (x, y) \in D_{C}} Loss(f_{\theta}(\phi_{\theta_{\phi}}(x)),y)$
>
> **Equation (5)**
>
> $ \dfrac{1}{N} \sum_{i \in D} \left | Loss(f(\phi(x_i)),y_i)  - Loss(\hat{f}\_{cpl}(\phi(x_i)),y_{cpl,i}) \right |   $
> $ \leq \dfrac{1}{N} \sum_{i \in D} (\left | Loss(f(\phi(x_i)),y_i)  - Loss(\hat{f}\_{y}(\phi(x_i)),y_{i}) \right | + \left | Loss(\hat{f}\_{y}(\phi(x_i)),y_{i}) - Loss(\hat{f}\_{cpl}(\phi(x_i)),y_{cpl,i}) \right | ) $
>
> **Equation (6)**
>
> The updated definition of equation (6) can be found in the attached PDF file of the global response. The first equation is corrected as $\hat{f}\_{ymap}^{j}(x_i) = \sum_{k = 1}^{K_{cpl}} \mathbb{I}_{j,k} \hat{f}\_{cpl}(x_i)$

---

### Decision · Program_Chairs · 2023-09-21

**Decision:**

Reject

**Comment:**

The paper introduces a novel approach called neural tangent kernel clustering-pseudo-labels (NTKCPL) that combines active learning with self-supervised learning to improve performance in the context of supervised learning with a lookhead strategy. The proposed method addresses the phase transition point and aims to extend the effective budget range for active learning. The paper includes theoretical and empirical analyses and presents experimental results on various datasets.

Most reviewers recognize the novelty of the proposed approach and the comprehensive analysis of the paper. Somehow they also point out that clarity and writing is the major weakness of the paper. The authors' rebuttal clarified most of the misunderstandings, making several reviewers increase their scores, placing the paper borderline acceptable under the big if that the long rebuttal contents can be merged into the main paper in a satisfactory manner. Somehow none of the reviewers strongly want to champion the paper, and even the more positive reviewers believe that the paper needs a significant revision to address the clarity issues raised during reviewing and discussion. Also, some reviewers suggest more comparison with other self-supervised + active learning possibilities to deepen the empirical study.